# Cell-intrinsic mTOR/LET-363 influences morphological aging of the ALM touch receptor neuron in *Caenorhabditis elegans*

**Sophia C. Whitley, Ruiling Zhong, Sophie Baumberger, Claire E. Richardson***

Department of Genetics, University of Wisconsin-Madison, Madison, Wisconsin, United States of America

* claire.richardson@wisc.edu

## Abstract

The mechanistic target of rapamycin (mTOR) promotes neuronal aging, but it remains unclear whether these effects arise from mTOR activity within neurons, other brain cell types, or peripheral tissues. Here, we tested the hypothesis that *Caenorhabditis elegans* mTOR/*let-363* functions cell-intrinsically in neurons during adulthood to promote age-related neuron morphological aging. We used a floxed *let-363* allele in combination with heat-shock-induced, pan-somatic Cre recombinase expression to generate pan-somatic, adult knockdown, and with a Cre-driver that expresses in a small subset of neurons, the Touch Receptor Neurons, to generate neuron-intrinsic knockdown. Adult-onset, pan-somatic knockdown of *let-363* did not robustly alter lifespan or neuron morphological aging. In contrast, neuron-specific *let-363* knockdown resulted in a reduction in one aspect of neuron morphological aging – ectopic neurite sprouting from the soma – without extending lifespan. Together, these findings suggest that mTOR/*let-363* can act cell-intrinsically within neurons to promote or potentiate an aspect of morphological aging. These results help clarify the potential cell-type specificity of mTOR's roles in neuronal aging and provide a foundation for defining the mechanisms through which mTOR intersects with neuron-intrinsic aging pathways.

## Introduction

Aging is a major driver of cognitive decline and neurodegenerative diseases including Alzheimer's disease (AD) [1]. As populational aging accelerates globally, understanding the molecular processes that govern neuronal aging has become increasingly important. Research across diverse species has identified genetic and environmental factors, especially those related to metabolic signaling, that influence age-associated neuronal dysfunction; the underlying mechanisms remain incompletely understood [1,2].

**Data availability statement:** All relevant data are within the manuscript and its Supporting information files.

**Funding:** This work was supported by the National Institutes of Health (https://www. nih.gov/) grant R35-GM154869 to C.E.R., the United States Department of Agriculture (https://www.usda.gov/) grant WIS05016 to C.E.R., the Sophomore Research Fellowship from UW-Madison to S.C.W., and the Hilldale Undergraduate/Faculty Reearch Fellowship from UW-Madison to S.C.W. The funders had no role in study design, data collection and analysis, decision to publish, or preparation of the manuscript.

**Competing interests:** No authors have competing interests.

The mechanistic target of rapamycin (mTOR) pathway is an evolutionary conserved signaling pathway that promotes growth, development, and aging [3–5]. mTOR is a central player in metabolic signaling; it is a core component of both the mTORC1 and mTORC2 complexes, which integrate nutrient and growth-factor signals to coordinate distinct sets of cellular processes [5–16]. In nutrient-replete conditions, mTOR promotes aging, in part by positively regulating protein synthesis and inhibiting autophagy; across diverse animals, reducing mTOR activity extends lifespan and health span [5].

Consistent with mTOR's role in promoting aging, inhibition of mTOR signaling has shown promising neuroprotective effects in animal models of cognitive aging and neurodegenerative diseases. Chronic rapamycin treatment extends lifespan and preserves cognitive performance during normal aging in wild-type mice [17–19]. Furthermore, in mouse models of AD, chronic rapamycin treatment reduces amyloid pathology and the emergence of cognitive deficits [20,21], and genetic reduction of mTOR leads to similar improvements [22]. These findings suggest that mTOR signaling contributes to neuronal dysfunction and that its inhibition can be protective.

Despite this promise, a challenge for realizing the therapeutic potential of mTOR inhibition in preserving neuronal health is that mTOR is required for diverse physiological processes. Strong, systemic inhibition of mTOR can cause harmful effects, including impaired nutrient sensing, metabolic dysregulation, and immunosuppression [23–29]. An important goal, therefore, is to identify the tissues and time windows in which mTOR activity promotes neuronal aging, enabling more targeted interventions with fewer pleotropic or antagonistic effects. Studies indicate that mTOR acts within neurons and non-cell-intrinsically to promote disease pathology in mouse models of neurodegenerative disease [22,30,31]. Where mTOR functions to promote neuron aging outside of neurodegenerative disease models is incompletely understood.

*C. elegans* provides a powerful experimental paradigm for addressing such questions. *C. elegans* neurons exhibit age-associated defects including sporadic morphological abnormalities, reduced synaptic vesicle accumulation at presynapses, and reduced neurotransmission [32–36]. *C. elegans* has a single mTOR homolog, *let-363,* which, as in other animals, is essential for larval development and promotes aging [37,38]. Somatic knockdown of mTOR signaling using auxin inducible degradation (AID) can cause a shortened lifespan but pan-neuronal knockdown of mTOR signaling extends lifespan [39–41]. These results indicate that *C. elegans* mTOR signaling has tissue-specific functions in the regulation of adult lifespan. Whether, when, and from where *let-363* impacts *C. elegans* neuron aging is not known.

In this study, we tested the hypothesis that *let-363* functions cell-intrinsically during adulthood to promote neuron morphological aging. By assessing neuron morphological aging and lifespan in conditional *let-363* knockdown manipulations, we show that *let-363* functions cell-intrinsically within the neuron to promote age-associated ectopic neurite sprouting from the soma without a concomitant increase in animal lifespan. Our results do not support the model that *let-363* functions specifically in adulthood to promote neuron morphological aging, which may indicate that *let-363* during development plays a role, but limitations of our study make this interpretation uncertain.

## Materials and methods

### *C. elegans* strain maintenance

All strains were grown on nematode growth medium (NGM) plates seeded with *E. coli* OP50 at 20 °C (S3 Table). Aged worms are defined by the number of days after the late L4 stage, "Day 0" of adulthood. Genotypes were confirmed by PCR (*let-363* and *heSi160*) or fluorescence (*zdIs5* and *wyIs592*). Genotyping primers used in this study are listed in S4 Table.

### Heat shock treatment

For all experiment using heat-shock to generate pan-somatic, adult *let-363* knockdown (Fig 1D-J), worms were grown to Day 0 of adulthood at 20 °C, then transferred to new NGM plates containing *E. coli* OP50. These plates were heat shocked for 1 hour at 34 °C, then transferred back to 20 °C until the next stage of the experiment. For the PVD dendritic arbor experiment to assess whether *heSi160[Phs > Cre]* can recombine *let-363(wy1706)* in neurons (Fig 1B-C), gravid worms were egg-laid overnight, then removed from the plates. The plates of eggs were heat-shocked for 2 hours at 34 °C, then transferred back to room temperature for further experiments.

### Lifespan assays

Lifespan assays were performed on a 6-cm NGM plates seeded with *E. coli* OP50. To prepare for the experiment, reproductively active adult hermaphrodites were egg-laid for 8 hours before being removed, ensuring an age-synchronized batch of animals. The resulting eggs were grown to Day 0 of adulthood. For each strain/condition being analyzed, worms were distributed across four replicate plates, each containing 25–35 worms. The strain genotypes were blinded to the experimenter.

The worms were examined every 1–3 days under a dissecting microscope to assess survival. While reproductively active, worms were regularly transferred to new NGM plates with *E. coli* OP50 to prevent progeny from maturing and being confused with the experimental cohort. Worms were censored from the analysis if they were accidently killed during transfers, were missing from the plate, or if the plate became contaminated.

### Neuron morphology experiments

Reproductively active adults were egg-laid for 8 hours, then removed to achieve an age-synchronized batch. Eggs were grown to Day 0 of adulthood, then transferred to new NGM plates containing *E. coli* OP50. For each strain being analyzed, worms were transferred to 3 replicate plates, each containing 15–20 worms. The experiment was conducted blinded to strain identity.

All genotype-by-treatment combinations were assayed in 3 biological replicates except for *let-363(1706)* -HS, which was included in just 2 biological replicates to increase feasibility. We considered the *let-363(wy1706)* -HS genotype-by-treatment to be the least useful comparison to *let-363(wy1706); heSi160[Phs > Cre]* +HS since both the genotype and treatment are different. All the worms in these morphology experiments were homozygous for the *zdIs5[Pmec-4 > GFP]* allele.

Every 2 days, worms were transferred to new NGM plates with *E. coli* OP50 to keep the cohort of adult worms distinct from their progeny. On adult day 12, slides were prepared with 30 μl of 5% agarose melted in M9 buffer, and 3 μl of 10 mM levamisole in M9 buffer. 6–10 worms were placed on a slide at a time. Neuron morphology was assessed using Nikon eclipse Ti2 microscope paired with a CSU W1 SoRa confocal scanner unit and a YOKOGAWA spinning disc unit with a Plan Apo VC 60xA/1.20 WI objective, with 488 nm laser at 40% laser power. Z-stack images were obtained using the same settings and were taken with 100 ms exposure time.

To establish a baseline for neuron morphology, a scoring scale was developed by assessing Day 12 *zdIs5; let-363; hesi160* -HS and *zdIs5; let-363* worms (Table 1).

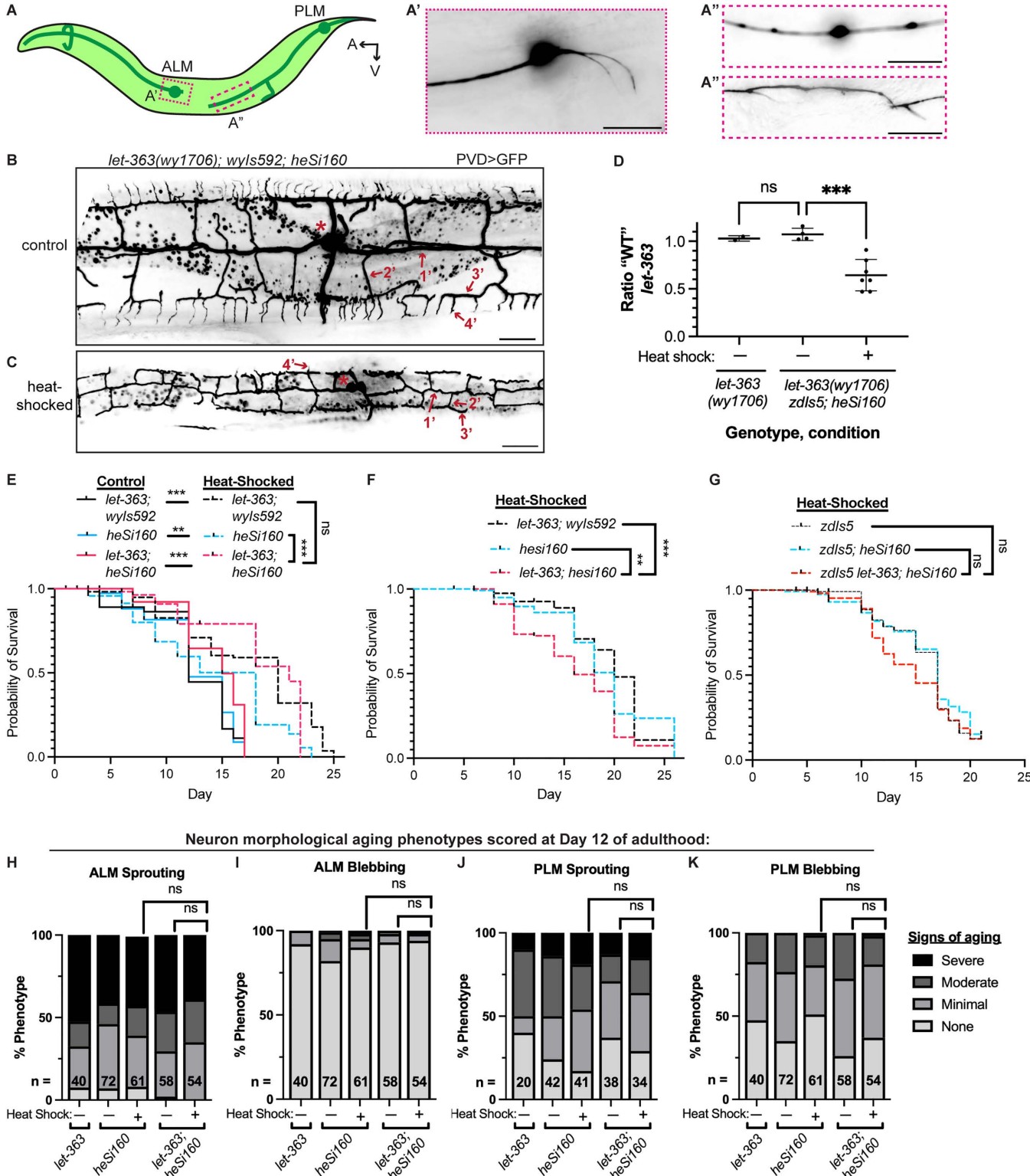

Fig 1. Pan-somatic, adult knockdown of *let-363* does not robustly alter lifespan or neuron morphological aging. (A) Schematic of the ALM and PLM neuron morphology in a wild-type young adult, with example microscopy images of the ALM sprouting phenotype (A'), PLM blebbing (A", upper), and PLM sprouting (A", lower). TRNs were visualized throughout this work with *zdIs5[P_{TRN}>GFP]*. The ALM blebbing phenotype is similar to PLM

blebbing (S1 Fig). Scale: 10 μm. **(B-C)** Representative images of the PVD neuron in *let-363(wy1706); wyIs592[P_{PVD} > GFP]; heSi160[P_{heatshock}>Cre]* adults that were not heat-shocked (B) versus an age-matched animal that was heat-shocked as an egg (C). Asterisks mark the soma, and example 1', 2', 3', and 4' dendrites of the arbor are indicated. Pan-somatic knockdown of *let-363* in (C) results in reduced dendritic complexity, including absence of most 4' dendrites, and a smaller animal. (D) ddPCR quantification of the fraction of non-Cre-recombined *let-363* alleles over the total alleles in Day 1 adults that were heat-shocked or not at Day 0 of adulthood. Points show technical replicates from one pooled population per genotype/treatment, plotted with mean ± SD. ***P < 0.001; ns: not significant; one-way ANOVA with Tukey post-test. **(E)** Lifespan curve of pan-somatic, adult knockdown animals (heat-shocked *let-363(wy1706); heSi160[P_{heatshock}>Cre]*) versus heat-shocked comparison genotypes and non-heat-shocked controls. *wyIs592[P_{PVD} > GFP, P_{AWC}>RFP]* is a neuron morphology marker and is not expected to impact lifespan. **(F)** Biological replicate lifespan experiment comparing heat-shocked *let-363; heSi160* to heat-shocked comparison strains. **(G)** Lifespan experiment comparing heat-shocked *zdIs5 let-363; heSi160* to heat-shocked comparison strains. ***P < 0.001, **P < 0.01; ns: not significant; pairwise Log-rank (Mantel-Cox) test with Holm-Bonferroni correction. **(H-K)** Quantification of neuron morphological aging at Day 12 of adulthood in pan-somatic, adult *let-363* knockdown animals versus comparison genotypes/conditions. All strains were homozygous for *zdIs5[P_{TRN}>GFP]*. A maximum of one ALM and one PLM neuron were scored per animal; the number of neurons scored for each phenotype is indicated in the bar. ns: not significant; Fisher's exact test.

For assessing PVD dendritic arbor morphology, gravid adults were egg-laid overnight, then removed, and the plates containing the eggs were either heat shocked or not. Worms were grown to Day 0 of adulthood, then transferred to new NGM plates containing *E. coli* OP50. Every 2 days, worms were transferred to new NGM plates with *E. coli* OP50 to prevent larvae from growing to full adults. On adult day 3, slides were prepared and animals were imaged as described above. Neuron dendritic arbor morphology was visualized and analyzed qualitatively.

Images were exported from Nikon as ND2 files and processed in FIJI.

## ddPCR

Digital Droplet PCR was chosen to validate the excision of *let-363* from the *C. elegans* genome due to the method's absolute quantification and high sensitivity. Primers and probes were designed as outlined in the Bio-Rad Droplet Digital PCR Applications Guide. Genomic DNA was prepared using the Zymo Research Quick-DNA Miniprep Plus Kit (Cat# D4068), following the "Solid Tissues" protocol. The final DNA concentration was measured using Microvolume UV-Vis Spectrophotometer (NanoDrop).

Primers and probes were ordered from IDT and stored as a 100 μL master mix of 180 μM primers/5 μM probe (S4 Table). The ddPCR reaction components for a 20 μL reaction were: 10 μL of 2x ddPCR supermix for probes, 1 μL of non-recombined *let-363* primer/FAM probe mix (final concentration: 900 nM primers/250 nM probe), 1 μL of genomic reference primer/HEX probe mix, and 8 μL of DNA template.

The optimal DNA concentration was determined to be on the scale of 0.1 ng/μL of the original concentration. This was determined by running a 10-fold dilution series of DNA concentration. We considered the 1-D droplet reader output plot,

**Table 1. Neuron morphological aging severity scale.**

| Morphology | No Signs of Aging: 1 | Minimal Signs of Aging: 2 | Moderate Signs of Aging: 3 | Severe Signs of Aging: 4 |
|---|---|---|---|---|
| ALM Sprouting | No sprout | Sprout less than or equal to length of cell body | • 2 sprouts less than or equal to length of cell body OR<br>• 1 sprout greater than length of cell body | 1 or 2 sprouts greater than length of cell body |
| ALM Blebbing | • No blebs<br>• 1 bleb | • 2–3 large blebs OR<br>• 1–6 small blebs | 3-6 large blebs and 0–6 small blebs | 6⁺ large blebs and 7⁺ small blebs |
| PLM Sprouting | None | 1-4 sprouts | 5-7 sprouts | 8⁺ sprouts |
| PLM Blebbing | • No blebs<br>• 1 bleb | • 2–6 large blebs OR<br>• 1–7 small blebs | • 1–10 large blebs and 1–7 small blebs OR<br>• 8–15 small blebs | • 11⁺ large blebs and 8⁺ small blebs OR<br>• 16⁺ small blebs |

which graphs fluorescence intensity vs droplet number, looking for a clear separation between positive and negative droplets. We also considered the FAM/HEX ratio that had the tightest confidence interval, which was found to be in the range of 100–500 copies/ 20 μL reaction.

Droplets were generated using a DG8 Cartridge for a QX200 Droplet Generator, and then the droplets were transferred to a 96-well PCR plate by gentle pipetting. The PCR plates were heat-sealed using the Bio-Rad's PX1 PCR Plate Sealer and pierceable foil heat seal. PCR amplification was done in the BioRad C1000 Touch Thermal Cycler, then the plate was placed in QX200 Droplet Reader, which analyzes each droplet individually using a two-color detection system (FAM and HEX). Positive droplets, which contain at least one copy of the target DNA molecule, will have increased fluorescence compared to negative droplets.

Data were analyzed using QuantaSoft Software, using the absolute quantification experiment. Fluorescent thresholds, where droplets above the threshold are scored as positive and droplets below are scored as negative, were specific to each 20 μL reaction and based on the separation of positive and negative droplets on the 1-D fluorescence intensity vs droplet number plot. The fluorescent thresholds established by the 1-D plot were applied to the 2-D plot (S1 Fig), and the ratios of target DNA to reference DNA were calculated by the QuantaSoft software and recorded. Droplet reader output was not considered if there were less than 10,000 droplets and the concentration of copies/ 20 μL was not in the 100–500 copy range.

## Statistical analysis

Statistical analysis was conducted using GraphPad Prism10. For lifespan assays, Log-rank tests were performed between each individual comparison, and the Holm-Bonferroni correction was used to account for multiple comparisons. For ddPCR, the mean of the technical replicates of the ddPCR was plotted, and the genotype/treatments were compared using a one-way ANOVA followed by Tukey post-test (Fig 1D) or a two-tailed t-test (Fig 2B). For neuronal morphology experiments, Fisher's exact test was performed between each individual comparison.

## Results and discussion

To test the hypothesis that mTOR functions cell-intrinsically in adulthood to promote neuron morphological aging, we used the *let-363(wy1706)* allele, in which the start codon and first eight exons are floxed [42]. We split the hypothesis into two sets of experiments: the first uses knockdown of *let-363* pan-somatically in adulthood, and the second uses neuron-specific *let-363* knockdown. To quantify neuron morphological aging, we used the touch receptor neurons (TRNs) ALM and PLM (Fig 1A). These are bilaterally symmetric neurons that comprise four of the six TRNs in the worm. These neurons have invariant morphology in young, wild-type adults, but they exhibit age-associated ectopic sprouting and axon blebbing that progressively increase in penetrance and severity (Fig 1A and S1 Fig) [32–34].

### Knockdown of mTOR/*let-363* in adulthood

To generate pan-somatic, adult knockdown, we used *heSi160[P_{heatshock}>Cre]*, an allele that expresses Cre pan-somatically upon heat-shock [43]. This allele has been shown to recombine floxed loci with good efficiency after heat-shock and shows low ectopic recombination in the absence of heat-shock [44].

First, we assessed whether the Cre-Lox recombination worked as expected. The *let-363(wy1706)* allele was previously studied in the PVD neuron, wherein knockout of *let-363* in the neuron's progenitor lineage strongly diminishes dendrite size and complexity [42]. We observed a similar reduction of dendrite growth in *let-363(wy1706); heSi160[P_{heatshock}>Cre]* adults that were heat-shocked as eggs (100% penetrance of reduced dendrite growth, n = 13 animals, 1 neuron per animal), but not in *let-363(wy1706); heSi160[P_{heatshock}>Cre]* adults that were not heat-shocked (0% penetrance, n = 16 animals) or in heat-shocked wild-type animals (0% penetrance, n = 14)(Fig 1B-C). These results confirm that the *let-363(wy1706); heSi160[P_{heatshock}>Cre]* strain functions as expected in the PVD neuron lineage.

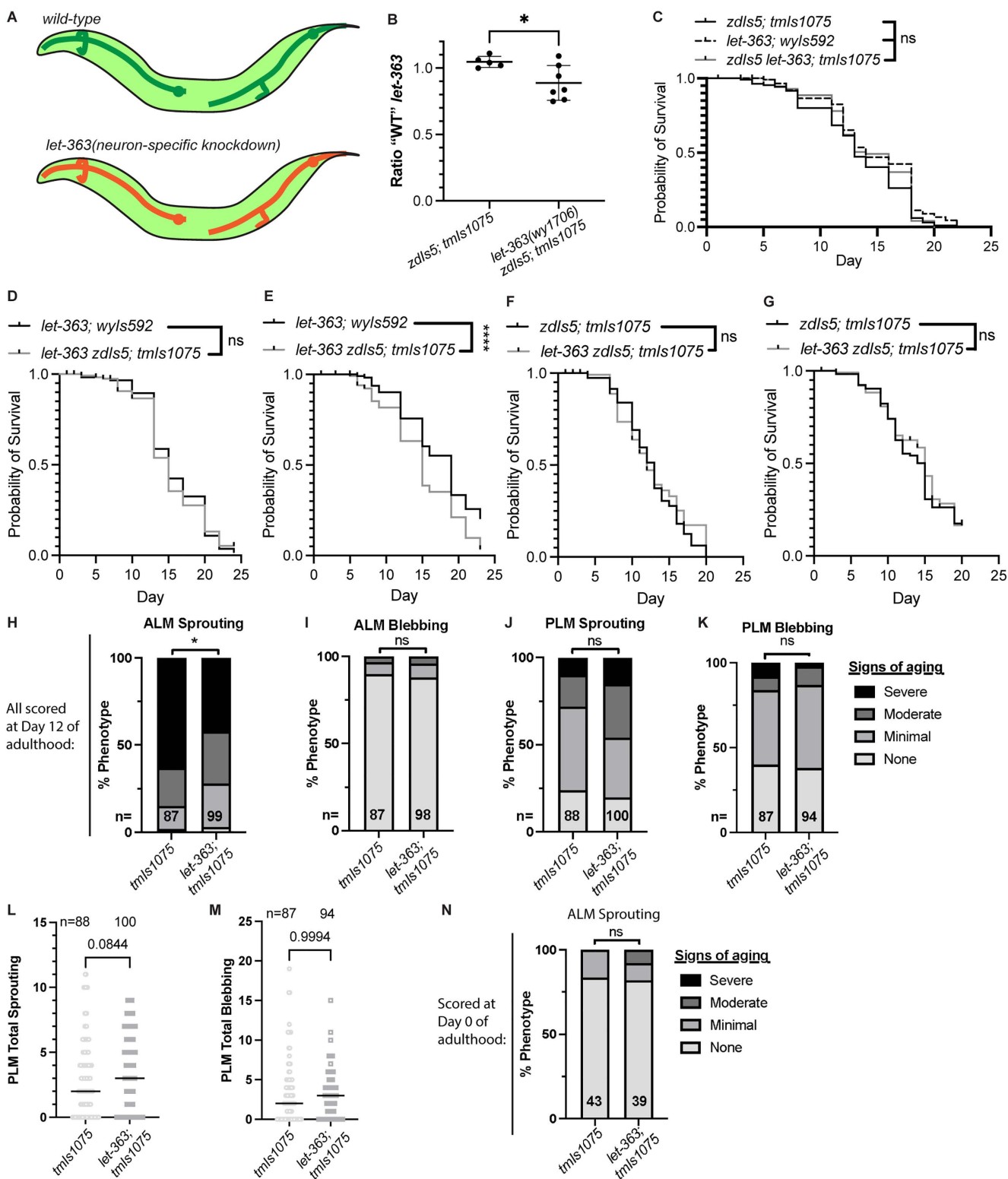

**Fig 2. Neuron-specific *let-363* knockdown reduces ectopic sprouting from the ALM soma without extending lifespan. (A)** Neuron-specific *let-363* knockdown was generated using *tmls1075[P_{TRN}>Cre]*, which is expected to be expressed shortly after TRN fate specification. **(B)** ddPCR quantification of the fraction of non-Cre-recombined *let-363* alleles over the total number of alleles in Day 1 adults. Points show technical replicates from one

pooled population per genotype, plotted with mean±SD. *P<0.05; two-tailed t-test. **(C-G)** Lifespan experiments comparing neuron-specific knockdown animals (*let-363(wy1706) zdIs5[$P_{TRN}$>GFP]; tmIs1075[$P_{TRN}$>Cre]*) to two control strains: *let-363(wy1706) wyIs592[$P_{PVD}$>GFP, $P_{AWC}$>RFP]* and *zdIs5[$P_{TRN}$>GFP]; tmIs1075[$P_{TRN}$>Cre]*. *wyIs592* is a neuron morphology marker and is not expected to impact lifespan. Each plot shows an independent biological replicate. ****P<0.0001; ns: not significant; Log-rank (Mantel-Cox) test. **(H-M)** Quantification of neuron morphological aging at Day 12 of adulthood in neuron-specific *let-363* knockdown animals versus comparison genotype. A maximum of one ALM and one PLM neuron were scored per animal; the number of neurons scored for each phenotype is indicated in the bar (H-K) or above **(L-M)**. **(G-J)** *P<0.05; ns: not significant; Fisher's exact test. **(K-L)** Mann-Whitney two-tailed t-test. Note that for PLM sprouting (I and K) and PLM blebbing **(J and L)**, the plots show alternate analyses of the same set of data. **(N)** Quantification of ALM sprouting at Day 0 of adulthood in neuron-specific *let-363* knockdown animals versus comparison genotype. The numbers in the bars indicate n; ns: not significant; Fisher's exact test.

We used ddPCR to assess the efficacy and specificity of Cre-mediated recombination in the *let-363(wy1706); heSi160[$P_{heatshock}$>Cre]* strain when the heat-shock is administered post-larval development (Fig 1D, S1 Table). In this experiment, Day 0 adults (several hundred age-matched animals spread across 3–6 plates per genotype/treatment) were heat-shocked for 1 hour at 34 °C or not heat-shocked, and all animals for each genotype/treatment were pooled for total genomic DNA extraction one day later. We detected no recombination of the *let-363* locus in *let-363(wy1706); heSi160[$P_{heatshock}$>Cre]* animals that were not heat-shocked (Fig 1D). In heat-shocked *let-363(wy1706); heSi160 [$P_{heatshock}$>Cre]* animals, we measured approximately 36% recombination (64% unrecombined; 95% C.I.: 49% – 80%) (Fig 1D). In young adults, about 66% of the genomes are in the germline (~1,500 nuclei per germline arm and ~300 self-sperm for ~3,300 germline genomes; there are ~1,460 somatic genomes: 959 somatic cells, several of which endoreduplicate their genomes) [45–47]. Therefore, *heSi160[$P_{heatshock}$>Cre]* works efficiently upon heat-shock and without detectable leakiness in the absence of heat-shock to recombine the *let-363(wy1706)* allele, generating a pan-somatic, adult *let-363* knockdown. Still, without a mTOR/LET-363 activity readout such as phosphorylation of S6K, we cannot rule out the possibility that LET-363 protein perdurance or systematic lack of recombination in specific cells/ tissues impact our results.

We next asked whether pan-somatic, adult knockdown of *let-363* impacts lifespan, using the 1 hour, 34 °C heat-shock at Day 0 of adulthood to generate knockdown after larval development in *let-363; heSi160[$P_{heatshock}$>Cre]* (Fig 1E). Of note, the heat-shock treatment itself caused a lifespan extension in the two comparison genotypes, *let-363(wy1706)* and *heSi160[$P_{heatshock}$>Cre],* increasing the median lifespan by 67% and 50%, respectively (Fig 1E, S2 Table). This is consistent with previous relthood [48–52]. The heat-shocked *let-363(wy1706); heSi160[$P_{heatshock}$>Cre]* strain showed a similar lifespan extension to the heat-shocked comparison strains, with a slightly longer lifespan compared to *heSi160[$P_{heatshock}$>Cre]* but no detectable difference compared to *let-363(wy1706)* (Fig 1E, S2 Table). In a biological replicate lifespan assay comparing the three heat-shocked strains, *let-363(wy1706); heSi160[$P_{heatshock}$>Cre]* showed a slightly shorter lifespan versus both comparison strains (Fig 1F, S2 Table). In an analogous lifespan experiment in which the same adult heat-shock treatment was administered to *zdIs5 let-363(wy1706); heSi160* alongside two control strains, we observed a subtle, but not statistically significant, reduction in lifespan with pan-somatic adult *let-363* depletion (mean lifespan 2 days, ~12% shorter, compared to controls)(Fig 1G, S2 Table). This lifespan reduction aligns with the results of Smith et al., 2023, who used auxin-inducible degradation (AID) to generate pan-somatic *let-363* knockdown at Day 1 of adulthood [39]. This lifespan reduction is not as robust in our paradigm, however (Fig 1E-G, S2 Table). The weaker effect here compared to Smith et al. may be due to perdurance of LET-363 protein after Cre-Lox excision, or a confounding effect of the heat-shock.

Next, we assessed the impact of pan-somatic, adult *let-363* knockdown on neuron morphological aging. We used the TRNs ALM and PLM, which are established to show "sprouting" and "blebbing" phenotypes that increase in penetrance and severity with age [32–34]. For both the ALM and PLM neurons, blebbing refers to swellings along the neurite (Fig 1A" and S1 Fig). For PLM, sprouting is also scored along the neurite (Fig 1A" and S1 Fig), whereas for the ALM neuron, sprouting refers to an ectopic neurite(s) emerging from the posterior of the soma (Fig 1A' and S1 Fig). As in the lifespan

experiments described above, we used a 1 hour, 34 °C heat-shock at Day 0 of adulthood to generate knockdown after larval development in *let-363; heSi160[P<sub>heatshock</sub>>Cre]*. Assessing neuron morphological aging at Day 12 of adulthood, we did not detect an impact of pan-somatic, adult *let-363* knockdown (Fig 1G-J). Somewhat surprisingly, we also did not detect a reduction in neuron morphological aging phenotypes from the heat-shock itself, despite the lifespan extension it generated (Fig 1G-J). We cannot exclude the possibility that the heat-shock itself, or pan-somatic adult *let-363* knockdown, provides a benefit for neuron morphological aging that manifests earlier or later in life.

### Neuron-intrinsic knockdown of mTOR/*let-363*

To generate neuron-specific *let-363* knockdown, we used *tmIs1075[P<sub>TRN</sub>>Cre]*, which is expected to express Cre early in the development of the TRNs (Fig 2A) [53]. We used ddPCR to assess *let-363* recombination in the *let-363(wy1706); tmIs1075[P<sub>TRN</sub>>Cre]* strain at Day 1 of adulthood (Fig 2B). The *let-363(wy1706); tmIs1075[P<sub>TRN</sub>>Cre]* strain showed a small but statistically significant amount of *let-363* excision compared to the control strain (Fig 2B). Due to the large variation between technical replicates (89% no recombination, 95% C.I.: 77% – 101%), it is unclear from these data whether more genomes than those of the six TRNs are recombined by *tmIs1075[P<sub>TRN</sub>>Cre]*(Fig 2B). We also note that without a cell-specific LET-363 activity reporter, we cannot rule out the possibility that *let-363* is not excised in the ALM and/or PLM neurons at some frequency, which would reduce the apparent impact of cell-intrinsic *let-363* on neuron morphological aging.

We inspected ALM and PLM morphology at Day 0 of adulthood (scored at the end of the L4 larval stage) qualitatively in 39 *let-363(wy1706); tmIs1075[P<sub>TRN</sub>>Cre]* animals, and we did not notice any abnormalities.

We next asked whether knockdown of *let-363* in the *let-363(wy1706); tmIs1075[P<sub>TRN</sub>>Cre]* strain impacts lifespan. We hypothesized that it would not, as the TRNs have no known role in regulating lifespan; however, if there were a lifespan effect, it would confound interpretation of a neuron morphological aging phenotype. We detected no robust change in lifespan of *let-363(wy1706); tmIs1075[P<sub>TRN</sub>>Cre]* compared to control strains (Fig 2C-G). In one three biological replicates, though, the *let-363(wy1706); tmIs1075[P<sub>TRN</sub>>Cre]* strain showed a mild but statistically significant decrease in lifespan compared to the *let-363(wy1706)* strain (Fig 2E).

Finally, we assessed the impact of neuron-intrinsic *let-363* knockdown on neuron morphological aging, again assessing the neurite sprouting and blebbing phenotypes in the ALM and PLM neurons at Day 12 of adulthood. Neuron-intrinsic knockdown of *let-363* resulted in a significant reduction in the severity of the ALM sprouting phenotype (P = 0.02), though it had no strong impact on the other phenotypes assessed (Fig 2H-M). We did not detect a difference in ALM sprouting in the *let-363(wy1706); tmIs1075[P<sub>TRN</sub>>Cre]* strain compared to the wild-type control at Day 0 of adulthood (Fig 2N). Together, these results suggests that mTOR acts within the ALM neuron to promote or potentiate the age-associated ectopic neurite sprouting from the soma. As the neuron-specific knockdown animals do not exhibit lifespan extension, this reduction of morphological aging is not an emergent property of extended lifespan.

Of note, the animals with neuronal *let-363* knockdown showed a trend for increased PLM sprouting, though it is not statistically significant (Fig 2J, P = 0.07, and 2L). It would be interesting to quantify that phenotype at an additional time-point(s) to determine whether the potential effect is robust.

In summary, our results suggest that mTOR cell-intrinsically promotes an aspect of neuron morphological aging – ectopic neurite sprouting from the soma – *in vivo*. Considering that we did not detect a decrease in neuron morphological aging in the pan-somatic, adult knockdown of *let-363,* which is expected to deplete *let-363* in neurons, we propose that the neuron-intrinsic *let-363* function during development and/or early adulthood may be important for its contribution to aging. Going forward, it would be interesting to determine how mTOR intersects with other factors known to regulate neuron aging cell-intrinsically [32,34,54–57]. In particular, a plausible hypothesis is that mTOR promotes neuron aging by negatively regulating TFEB/HLH-30 activity, as HLH-30 was recently shown to be instructive for limiting aspects of neuron morphological aging [57].

## Supporting information

**S1 Fig. Example images of the neuron morphology phenotypes scored.** TRNs were visualized with *zdIs5[P$_{TRN}$>GFP]* at Day 12 of adulthood in strain PBT206 *zdIs5 let-363(wy1706); tmIs1075*. Scale: 10 μm.
(TIF)

**S2 Fig. Example of 2-D droplet reader output plot for ddPCR to show separation of positive and negative droplets.**
(TIF)

**S1 Table. Raw data for ddPCR.**
(XLSX)

**S2 Table. Raw data for lifespan assay.**
(XLSX)

**S3 Table. *Caenorhabditis elegans* strains used in this paper.**
(XLSX)

**S4 Table. Primers used in this paper.**
(XLSX)

## Acknowledgments

Some *C. elegans* strains were provided by the Caenorhabditis Genetics Center (CGC). Additional strains were provided by the National BioResource Project, and the Kang Shen lab, Stanford University. We thank Sherlyn Wijaya, Manuel Alvarez, and other members of the Richardson lab for discussions. We thank Dan Stevens from the Biotechnology Center at the University of Wisconsin – Madison for assistance in the experimental design, training and data analysis for ddPCR. We thank the CALS Honors in Research Program for helping develop research skills and monitoring progress for S.C.W.

## Author contributions

**Conceptualization:** Claire E. Richardson.

**Data curation:** Sophia C. Whitley, Claire E. Richardson.

**Formal analysis:** Sophia C. Whitley, Ruiling Zhong, Claire E. Richardson.

**Funding acquisition:** Sophia C. Whitley, Claire E. Richardson.

**Investigation:** Sophia C. Whitley, Ruiling Zhong, Sophie Baumberger, Claire E. Richardson.

**Methodology:** Sophia C. Whitley, Ruiling Zhong, Claire E. Richardson.

**Project administration:** Claire E. Richardson.

**Resources:** Claire E. Richardson.

**Supervision:** Claire E. Richardson.

**Validation:** Sophia C. Whitley, Ruiling Zhong, Sophie Baumberger.

**Visualization:** Sophia C. Whitley, Ruiling Zhong, Claire E. Richardson.

**Writing – original draft:** Sophia C. Whitley, Claire E. Richardson.

**Writing – review & editing:** Sophia C. Whitley, Ruiling Zhong, Claire E. Richardson.

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
