## [Decision Letter · Decision Letter 0]

21 Dec 2025

*Caenorhabditis elegans*PLOS One

Dear Dr.  Richardson,

Thank you for submitting your manuscript to PLOS ONE. After careful consideration, we feel that it has merit but does not fully meet PLOS ONE’s publication criteria as it currently stands. Therefore, we invite you to submit a revised version of the manuscript that addresses the points raised during the review process.

We look forward to receiving your revised manuscript.

Kind regards,

Erik A. Lundquist

Academic Editor

PLOS One

Journal Requirements:

Additional Editor Comments:

Overall the reviewers thought this was a strong manuscript and an important contribution. The authors should address the points raised by the reviewers and possibly add new data and/or figure panels, or strongly justify why not. In particular, comments about showing images of phenotypes in Table 1 and numbers of replicates in some of the aging experiments (two are shown and the reviewers suggest 3). Also, the "data not shown" should be shown in supplemental data or not referenced.

Reviewer's Responses to Questions

**Comments to the Author**

1. Is the manuscript technically sound, and do the data support the conclusions?

Reviewer #1: Yes

Reviewer #2: Yes

2. Has the statistical analysis been performed appropriately and rigorously?

Reviewer #1: Yes

Reviewer #2: Yes

3. Have the authors made all data underlying the findings in their manuscript fully available?

Reviewer #1: Yes

Reviewer #2: Yes

4. Is the manuscript presented in an intelligible fashion and written in standard English?

Reviewer #1: Yes

Reviewer #2: Yes

Reviewer #1: In this manuscript, Whitley et al investigate the role of mTOR in the age-related changes observed in the ALM touch receptor neuron in C. elegans. The manuscript flows logically and experiments are well explained. Though incremental, the contributions from this paper help address one of the likely multiple mechanisms behind the age-related changes to the morphology of the ALM neurons.

Below are some major points that, upon addressed, will significantly strengthen the manuscript:

1) Fig 1 would strongly benefit from images of wilt-type ALM and PLM neurons, and of all phenotypes associated with Table 1 (even if in supplemental). This would help those who are not so familiar with these neurons comprehend the phenotype better. As a side note, is there a reason for the ‘data not shown’ other than just choosing not to show the data (despite unlimited space in supplemental figures)?

2) Phenotypes (blebbing, sprouting) are binned, which could mask more subtle but important phenotypes. For sprouting, I would recommend plotting numbers of sprouts per neurons (which could perhaps even show a significant phenotype in Fig 2I)

3) Lifespan assays in the manuscript are done in two replicates, though three replicates are more often the standard in the field. Hence, a third rep for each lifespan experiment would be highly suggested. In fact, for the cases where discrepancies were found between repetitions (i.e., Fig 1E and F and Fig 2D and F), a third repetition would aid in understanding which lifespan assay trend is to be believed.

4) Can the authors provide some sort of mTOR activity readout (i.e. p4EBP, pS6K) in vivo to show that mTOR is in fact active in the ALM and PLM neurons? This would exclude the possibility that mTOR is also acting in cells other than ALM/PLM (even in the 'specific' knockdowns). In addition, a time-dependent decrease in mTOR activity in ALM/PLM neurons would correlate with the author's hypothesis that the phenotypes observed are physiologically relevant to aging.

Minor points:

1) The first paragraph in the introduction needs references (it’s a whole paragraph without any references).

2) In the sentence, “In one biological replicate, though, the let-363(wy1706); tmIs1075[PTRN>Cre] strain showed a mild but statistically significant decrease in lifespan compared to the let-363(wy1706) strain (Fig 2F)”, I believe the authors mean Fig 2D.

Reviewer #2: Review:

PONE-D-25-63170

The work focuses on the question of whether mTor functions in a neuron-specific manner (as opposed to other cell types) to control aging / They a generated post-developmental knockdown of mTor (let-363) (which is in the mTor1 and mTor2 complex) in the touch neurons. The allele is lethal, so constructing a post-development knockout, that is specific to touch neurons, is important for the study and is a strength of the study. This study is also important to address the discrepancy of mTor knockdown and observed phenotypes: somatic knockdown results in a shortened lifespan, but pan neuronal knockdown extends the lifespan.

Introduction:

Well written and sets up the study nicely. One fundamental question, perhaps outside of the study but important to acknowledge and consider, is that the mTorc1 and mTorc2 complexes in C. elegans and mammals have been shown to regulate specific and sometime separate outcomes neurons. A let-363 knockdown will alter both complexes and will perhaps make the results harder to interpret.

Methods:

I am curious to know why the let-363(1706) worms were only included in 2 replicates while the rest were included in 3.

This sentence could use clarity: “Every 2 days, worms were transferred to new NGM plates with E. coli OP50 to prevent larvae from growing to full adults.” What I think you mean is that the adult worms were transferred so they would not be contaminated by the growing prodigy and thus you would be able to track the original population.

Very nice chart describing how you are quantifying your phenotypes in the worms.

I think it would be important to clarify throughout that “post-developmental knockout” in this case means throughout. I had trouble following the groups and when they were heat shocked.

I note you scored the neurons at day 12. Did you look at them earlier to make sure neuronal development was not impaired I the groups heat shocked as eggs?

Results and discussion:

The authors split the hypothesis into 2 sets of experiments:

1. Let-363 knockdown pan-somatically in adulthood

2. Neuron specific let-363 knockdown.

You mention that the neurons have invariant morphology in young, wild-type adults. Does this changes with either of the knockdowns that you have described?

Figure 1

This looks at pan-somatic knockdown of let-363 (using let-363(wy1706) and it does not robustly alter lifespan or neuron morphological again. This figure legend could be read as the animals were heat shocked at day 1 of adulthood, but the methods suggest that you heat shocked eggs. I do see that you clarified in (B), but I would recommend clarifying the ages of the worms and when they are heat shocked throughout.

I see that neuron morphological aging is quantified at day 12 in the figure legend. It might be useful to indicate that in the figure for easy.

When you knockdown let-363 post-developmentally, when is the heat shock occuring? I would keep this clear throughout for ease. I can see day 0 buried in the paragraph, but it would be useful to keep it clear.

“This lifespan reduction is not robust in our paradigm, however, as it occurred in only one of two biological replicates” It might be useful to run another replicate or two to tease this out. Is there a reason you did not?

“We cannot exclude the possibility that the heat-shock, or post developmental pan-somatic let-363 knockdown, provides a benefit for neuron morphological aging that manifests earlier or later in life.” It could also be off target effects because you saw this in several of the heat shock groups and you mentioned that other labs see the same thing.

Neuron specific knockdown. No robust change in lifespan, as expected. Neuron intrinsic knockdown of let-363 resulted in significant reduction in ALM blebbing. It seems like you did not get robust knockdown using this method. Perhaps that could be a reason there are not more changes in phenotypes.

Interpretation of results is reasonable. References are good.

Overall a very good manuscript, I recommend minor edits for clarity.

**Do you want your identity to be public for this peer review?** For information about this choice, including consent withdrawal, please see our For information about this choice, including consent withdrawal, please see our Privacy Policy .

Reviewer #1: No

Reviewer #2: No

---

## [Author Response · Author response to Decision Letter 1]

26 Feb 2026

Editor Comments:

Overall the reviewers thought this was a strong manuscript and an important contribution. The authors should address the points raised by the reviewers and possibly add new data and/or figure panels, or strongly justify why not. In particular, comments about showing images of phenotypes in Table 1 and numbers of replicates in some of the aging experiments (two are shown and the reviewers suggest 3). Also, the "data not shown" should be shown in supplemental data or not referenced.

We addressed all of the reviewers points as described below, and we added the following additional data: Fig 1G, Fig 2C, Fig 2L-N, S1 Fig.

Reviewer #1: In this manuscript, Whitley et al investigate the role of mTOR in the age-related changes observed in the ALM touch receptor neuron in C. elegans. The manuscript flows logically and experiments are well explained. Though incremental, the contributions from this paper help address one of the likely multiple mechanisms behind the age-related changes to the morphology of the ALM neurons.

Below are some major points that, upon addressed, will significantly strengthen the manuscript:

1) Fig 1 would strongly benefit from images of wilt-type ALM and PLM neurons, and of all phenotypes associated with Table 1 (even if in supplemental). This would help those who are not so familiar with these neurons comprehend the phenotype better. As a side note, is there a reason for the ‘data not shown’ other than just choosing not to show the data (despite unlimited space in supplemental figures)?

We added Supplementary Figure 1, which shows example microscopy images of neurons in most phenotypic categories. Concerning the ‘data not shown,’ we did not include images of ALM neuron blebbing originally because we thought they seemed redundant with the images of PLM blebbing. We appreciate the reviewer’s point that with no space limits, it would be nice to include the images. They are now included in S1 Fig.

2) Phenotypes (blebbing, sprouting) are binned, which could mask more subtle but important phenotypes. For sprouting, I would recommend plotting numbers of sprouts per neurons (which could perhaps even show a significant phenotype in Fig 2I)

We added two plots for the neuron-specific let-363 knockdown, one showing the number of blebs and the other showing the number of sprouts per PLM neuron (Fig 2L-M). In quantifying neuron morphology with pan-somatic, adult knockdown (Figure 1), we only scored each neuron categorically, so we cannot add the same alternate analysis to Figure 1.

3) Lifespan assays in the manuscript are done in two replicates, though three replicates are more often the standard in the field. Hence, a third rep for each lifespan experiment would be highly suggested. In fact, for the cases where discrepancies were found between repetitions (i.e., Fig 1E and F and Fig 2D and F), a third repetition would aid in understanding which lifespan assay trend is to be believed.

We added a third repetition for each lifespan (Fig 1G and Fig 2C).

4) Can the authors provide some sort of mTOR activity readout (i.e. p4EBP, pS6K) in vivo to show that mTOR is in fact active in the ALM and PLM neurons? This would exclude the possibility that mTOR is also acting in cells other than ALM/PLM (even in the 'specific' knockdowns). In addition, a time-dependent decrease in mTOR activity in ALM/PLM neurons would correlate with the author's hypothesis that the phenotypes observed are physiologically relevant to aging.

This is an excellent point. We did not have the manpower to execute this, but we added text to the Results and Discussion section to note that a readout along these lines would be necessary to “rule out the possibility that LET-363 perdurance or systematic lack of recombination in specific cells/tissues impact our results” (lines 326-329). Later in the Results and Discussion, we also added this sentence: “We also note that without a cell-specific LET-363 activity reporter, we cannot rule out the possibility that let-363 is not excised in the ALM and/or PLM neurons at some frequency, which would reduce the apparent impact of cell-intrinsic let-363 on neuron morphological aging” (lines 390-393).

Minor points:

1) The first paragraph in the introduction needs references (it’s a whole paragraph without any references).

We added two references to that first paragraph.

2) In the sentence, “In one biological replicate, though, the let-363(wy1706); tmIs1075[PTRN>Cre] strain showed a mild but statistically significant decrease in lifespan compared to the let-363(wy1706) strain (Fig 2F)”, I believe the authors mean Fig 2D.

Yes, thank you. We fixed this.

Reviewer #2: Review:

PONE-D-25-63170

The work focuses on the question of whether mTor functions in a neuron-specific manner (as opposed to other cell types) to control aging / They a generated post-developmental knockdown of mTor (let-363) (which is in the mTor1 and mTor2 complex) in the touch neurons. The allele is lethal, so constructing a post-development knockout, that is specific to touch neurons, is important for the study and is a strength of the study. This study is also important to address the discrepancy of mTor knockdown and observed phenotypes: somatic knockdown results in a shortened lifespan, but pan neuronal knockdown extends the lifespan.

Introduction:

Well written and sets up the study nicely. One fundamental question, perhaps outside of the study but important to acknowledge and consider, is that the mTorc1 and mTorc2 complexes in C. elegans and mammals have been shown to regulate specific and sometime separate outcomes neurons. A let-363 knockdown will alter both complexes and will perhaps make the results harder to interpret.

While our study does not experimentally distinguish between mTORC1 and mTORC2, this is a good point that the existence of the two complexes that have distinct functionalities is important background information. We added an acknowledgement that mTOR functions in both the mTORC1 and mTORC2 complexes to the introduction, along with a handful of references that address the distinct functions of the two complexes in C. elegans (lines 55-56).

Methods:

I am curious to know why the let-363(1706) worms were only included in 2 replicates while the rest were included in 3.

Other readers would probably wonder about that as well, so we added the following explanation: “All genotype-by-treatment combinations were assayed in 3 biological replicates except for let-363(1706) -HS, which was included in just 2 biological replicates to increase feasibility. We considered the let-363(wy1706) -HS genotype-by-treatment to be the least useful comparison to let-363(wy1706); heSi160[Phs>Cre] +HS since both the genotype and treatment are different” (lines 149-152).

This sentence could use clarity: “Every 2 days, worms were transferred to new NGM plates with E. coli OP50 to prevent larvae from growing to full adults.” What I think you mean is that the adult worms were transferred so they would not be contaminated by the growing prodigy and thus you would be able to track the original population.

Yes, thank you for pointing this out, the original wording was confusing. We clarified the wording as follows: “Every 2 days, worms were transferred to new NGM plates with E. coli OP50 to keep the cohort of adult worms distinct from their progeny” (lines 155-156).

Very nice chart describing how you are quantifying your phenotypes in the worms.

I think it would be important to clarify throughout that “post-developmental knockout” in this case means throughout. I had trouble following the groups and when they were heat shocked.

We adjusted the wording throughout the manuscript so that we consistently describe that group as “pan-somatic, adult knockdown.”

I note you scored the neurons at day 12. Did you look at them earlier to make sure neuronal development was not impaired I the groups heat shocked as eggs?

The pan-somatic, adult let-363 knockdowns used to score neuron morphological aging at Day 12 were performed at Day 0 of adulthood (the same day that the L4 larval stage ends). Whether an earlier heat-shock would cause a developmental defect in the ALM and PLM neurons is beyond the scope of the question we aim to address with this study. We guess that this reviewer may have mixed up our pan-somatic, adult let-363 manipulations with the validation experiment using the PVD dendrite morphology due to our unclear wording (Fig 1B-C). To address this point, therefore, we adjusted the wording in the methods section with the intention of clarifying the different treatments for the validation experiment and the hypothesis-testing experiment. That section now reads:

“For all experiment using heat-shock to generate pan-somatic, adult let-363 knockdown (Fig 1D-J), worms were grown to Day 0 of adulthood at 20 °C, then transferred to new NGM plates containing E. coli OP50. These plates were heat shocked for 1 hour at 34 °C, then transferred back to 20 °C until the next stage of the experiment. For the PVD dendritic arbor experiment to assess whether heSi160[Phs>Cre] can recombine let-363(wy1706) in neurons (Fig 1B-C), gravid worms were egg-laid overnight, then removed from the plates. The plates of eggs were heat-shocked for 2 hours at 34 °C, then transferred back to room temperature for further experiments” (lines 118-124).

We also adjusted the wording describing these manipulations in the Results and Discussion section. The wording adjustments were in these sections:

“We used ddPCR to assess the efficacy and specificity of Cre-mediated recombination in the let-363(wy1706); heSi160[Pheatshock>Cre] strain when the heat-shock is administered post-larval development (Fig 1D, S1 Table)” (lines 283-285).

“We next asked whether pan-somatic, adult knockdown of let-363 impacts lifespan, using the 1 hour, 34 �C heat-shock at Day 0 of adulthood to generate knockdown after larval development in let-363; heSi160[Pheatshock>Cre] (Fig 1E)” (lines 306-308).

“As in the lifespan experiments described above, we used a 1 hour, 34 �C heat-shock at Day 0 of adulthood to generate knockdown after larval development in let-363; heSi160[Pheatshock>Cre]” (lines 336-338).

Results and discussion:

The authors split the hypothesis into 2 sets of experiments:

1. Let-363 knockdown pan-somatically in adulthood

2. Neuron specific let-363 knockdown.

You mention that the neurons have invariant morphology in young, wild-type adults. Does this changes with either of the knockdowns that you have described?

To address this question for the neuron-specific let-363 knockdown, we added the following text: “We inspected ALM and PLM morphology at Day 0 of adulthood (scored at the end of the L4 larval stage) qualitatively in 39 let-363(wy1706); tmIs1075[PTRN>Cre] animals, and we did not notice any abnormalities” (lines 387-389). We also added Fig 2N and the following text: “We did not detect a difference in ALM sprouting in the let-363(wy1706); tmIs1075[PTRN>Cre] strain compared to the wild-type control at Day 0 of adulthood (Fig 2N)” (lines 399-401).

This question does not seem relevant for the pan-somatic, adult let-363 knockdown, as the heat-shock treatment to excise let-363 was performed in young adults.

Figure 1

This looks at pan-somatic knockdown of let-363 (using let-363(wy1706) and it does not robustly alter lifespan or neuron morphological again. This figure legend could be read as the animals were heat shocked at day 1 of adulthood, but the methods suggest that you heat shocked eggs. I do see that you clarified in (B), but I would recommend clarifying the ages of the worms and when they are heat shocked throughout.

We addressed this point as described above.

I see that neuron morphological aging is quantified at day 12 in the figure legend. It might be useful to indicate that in the figure for easy.

We added a label of the animal age to the figures.

When you knockdown let-363 post-developmentally, when is the heat shock occuring? I would keep this clear throughout for ease. I can see day 0 buried in the paragraph, but it would be useful to keep it clear.

We addressed this point as described above.

“This lifespan reduction is not robust in our paradigm, however, as it occurred in only one of two biological replicates” It might be useful to run another replicate or two to tease this out. Is there a reason you did not?

We performed a third lifespan experiment and added those data (Fig 2G). The third replicate gave an intermediate result compared to the original two – there was a trend of a reduction in lifespan with the pan-somatic, adult let-363 knockdown that was not statistically significant. We adjusted the text in the figure caption and Results and Discussion section to include this new experiment.

“We cannot exclude the possibility that the heat-shock, or post developmental pan-somatic let-363 knockdown, provides a benefit for neuron morphological aging that manifests earlier or later in life.” It could also be off target effects because you saw this in several of the heat shock groups and you mentioned that other labs see the same thing.

We are not certain what this comment means, but we think it might be suggesting that there could be off target effects of the heat shock. We had intended to articulate that notion in the original sentence. To clarify the sentence, we added “itself” after “heat-shock” (line 342).

Neuron specific knockdown. No robust change in lifespan, as expected. Neuron intrinsic knockdown of let-363 resulted in significant reduction in ALM blebbing. It seems like you did not get robust knockdown using this method. Perhaps that could be a reason there are not more changes in phenotypes.

That is possible. We added a sentence noting that: “We also note that without a cell-specific LET-363 activity reporter, we cannot rule out the possibility that let-363 is not excised in the ALM and/or PLM neurons at some frequency, which would reduce the apparent impact of cell-intrinsic let-363 on neuron morphological aging” (lines 366-369).

Interpretation of results is reasonable. References are good.

Overall a very good manuscript, I recommend minor edits for clarity.

---

## [Editor Report · Decision Letter 1]

9 Mar 2026

Cell-intrinsic mTOR/LET-363 influences morphological aging of the ALM touch receptor neuron in *Caenorhabditis elegans*

PONE-D-25-63170R1

Dear Dr. Richardson,

We’re pleased to inform you that your manuscript has been judged scientifically suitable for publication and will be formally accepted for publication once it meets all outstanding technical requirements.

Kind regards,

Erik A. Lundquist

Academic Editor

PLOS One
---

## [Editor Report · Acceptance letter]

PONE-D-25-63170R1

PLOS One

Dear Dr. Richardson,

I'm pleased to inform you that your manuscript has been deemed suitable for publication in PLOS One. Congratulations! Your manuscript is now being handed over to our production team.

Kind regards,

on behalf of

Dr. Erik A. Lundquist

Academic Editor

PLOS One